# Survival Analysis of 3 Different Age Groups and Prognostic Factors among 402 Patients with Skeletal High-Grade Osteosarcoma. Real World Data from a Single Tertiary Sarcoma Center

**DOI:** 10.3390/cancers13030486

**Published:** 2021-01-27

**Authors:** Richard E. Evenhuis, Ibtissam Acem, Anja J. Rueten-Budde, Diederik S. A. Karis, Marta Fiocco, Desiree M. J. Dorleijn, Frank M. Speetjens, Jakob Anninga, Hans Gelderblom, Michiel A. J. van de Sande

**Affiliations:** 1Department of Orthopedic Surgery, Leiden University Medical Center, 2300RC Leiden, The Netherlands; i.acem@lumc.nl (I.A.); d.s.a.karis@lumc.nl (D.S.A.K.); D.M.J.Dorleijn@lumc.nl (D.M.J.D.); M.A.J.van_de_Sande@lumc.nl (M.A.J.v.d.S.); 2Department of Surgical Oncology and Gastrointestinal Surgery, Erasmus MC Cancer Institute, 3000CB Rotterdam, The Netherlands; 3Department of Biomedical Data Science, Section Medical Statistics and Bioinformatics, Mathematical Institute Leiden University, 2300RC Leiden, The Netherlands; a.j.ruten-budde@math.leidenuniv.nl (A.J.R.-B.); m.fiocco@math.leidenuniv.nl (M.F.); 4Department of Medical Oncology, Leiden University Medical Center, 2300RC Leiden, The Netherlands; F.M.Speetjens@lumc.nl (F.M.S.); A.J.Gelderblom@lumc.nl (H.G.); 5Princess Máxima Center for Pediatric Oncology, 3720AC Utrecht, The Netherlands; J.K.Anninga@prinsesmaximacentrum.nl

**Keywords:** osteosarcoma, survival, prognosis, age groups, chemotherapy, metastasis, local recurrence

## Abstract

**Simple Summary:**

Age is one of many prognostic factors for overall survival in patients with skeletal osteosarcoma. This retrospective study provides an overview of survival in patients with high-grade osteosarcoma in different age groups. It shows prognostic variables for survival and local control among the overall cohort. In this study, in which 402 patients with skeletal high-grade osteosarcoma were included, poor survival was associated with increasing age. Age groups, tumor size, poor histopathological response, distant metastasis at presentation, and local recurrence were independent prognostic factors associated to overall survival and event-free survival. Differences in outcome among different age groups can be partially explained by patient characteristics and treatment characteristics.

**Abstract:**

Age is a known prognostic factor for many sarcoma subtypes, however in the literature there are limited data on the different risk profiles of different age groups for osteosarcoma survival. This study aims to provide an overview of survival in patients with high-grade osteosarcoma in different age groups and prognostic variables for survival and local control among the entire cohort. In this single center retrospective cohort study, 402 patients with skeletal high-grade osteosarcoma were diagnosed and treated with curative intent between 1978 and 2017 at the Leiden University Medical Center (LUMC). Prognostic factors for survival were analyzed using a Cox proportional hazard model. In this study poor overall survival (OS) and event-free survival (EFS) were associated with increasing age. Age groups, tumor size, poor histopathological response, distant metastasis (DM) at presentation and local recurrence (LR) were important independent prognostic factors influencing OS and EFS. Differences in outcome among different age groups can be partially explained by patient and treatment characteristics.

## 1. Introduction

High-grade conventional osteosarcoma is a primary malignant bone tumor that has a bimodal distribution curve. The first peak is at the age of puberty and adolescence, the second curve arises after the age of 40 [1,2]. Despite being a rare disease (prevalence of 3–4 cases per million per year [3,4]), osteosarcoma is the most common primary malignant bone tumor. It continues to be a high risk malignancy and has one of the highest mortality rates of any type of cancer diagnosed around puberty [5]. Before the introduction of chemotherapy in the 1980’s, survival for patients with high-grade osteosarcoma was poor with survival probabilities as low as less than 20% [3]. After the introduction of chemotherapy, the overall survival (OS) increased to an average of 60% [3,6,7].

Multiple studies conclude more favorable survival probabilities in pediatric patients compared with adolescent and young adults (AYA) or older adults [8,9,10]. In contrast, some studies stated that no differences in survival were found between pediatric patients and older adults [11,12]. The variation in survival probabilities among age groups might be due to differences in tumor characteristics, chemotherapy regimens, pathohistological response, or different patient characteristics [9,13,14,15,16,17,18]. 

The aim of this single center retrospective study is to provide an overview of survival outcome within three age groups (pediatric, AYA, adult) and for the total cohort. The second aim is to identify prognostic factors for OS and event-free survival (EFS) in patients with high-grade osteosarcoma. 

## 2. Methods

### 2.1. Design, Setting, Data Source, Participants

This observational retrospective cohort study was performed at the Leiden University Medical Center (LUMC) in the Netherlands between 1978 and 2017. All consecutive patients diagnosed with histologically proven high-grade osteosarcoma treated with curative intent that met inclusion criteria were included. Patients with a skeletal high-grade primary osteosarcoma, treated with curative intent using (neo)adjuvant chemotherapy and surgery, were eligible for this study. Patients were excluded if they were diagnosed with, a low grade (parosteal) or intermediate grade osteosarcoma (peri-osteal), had a secondary osteosarcoma (i.e., radiation-induced), received a treatment with palliative intent, if data about surgery or chemotherapy were missing, or when the tumor location was facial or extra-skeletal. Patients with metastasis at presentation were eligible when curative intent was set at start of the treatment including planned metastasectomy. High-grade osteosarcoma consists of conventional osteosarcoma (osteoblastic, chondroblastic and fibroblastic), small cell and telangiectatic osteosarcoma. Apart from these subgroups, the WHO distinguishes high-grade surface osteosarcoma and secondary osteosarcoma as other types of high-grade osteosarcoma. This study was approved by the medical ethical committee of the LUMC as no patients were approached and data were handled anonymously. The approval code is G18.065/SH/gk. The used data comprised real world data.

### 2.2. Variables

Baseline variables were age, sex, location and size of the tumor and distant metastasis (DM) at presentation. Treatment data include LR, surgical margin, type of resection and response to chemotherapy. Patients were categorized into one of three age groups (children 0–<16, AYA 16–<40, older adults ≥ 40). Location of the primary tumor was defined as extremity (upper or lower extremity) or axial (tumors of the chest including ribs, spine or pelvis). The size of the primary tumor was divided according to the American Joint Committee on Cancer (AJCC) into small (≤8 cm) or large (>8 cm) [19]. Radical resection was defined as a wide radical resection with both macroscopic as microscopic surgical margins free of tumor and the entire dissection performed through healthy tissues. Marginal surgical margin was defined as a dissection that extended into or through the reactive zone that surrounds the tumor. Irradical or intralesional margin was defined as entering the tumor at any point during surgery [20]. 

The type of resection was divided into 3 subgroups; (1). reconstruction with an allo- or autograft, prosthesis or rotationplasty; (2). amputation of the affected limb or exarticulation of the joint without reconstruction; (3). resection that consisted of local resection, en-bloc resection or hemipelvectomy without reconstruction. The protocolized planned chemotherapy was either an intentional treatment with (Methotrexate, Doxorubicin, Cisplatin (MAP) or with Doxorubicin, Cisplatin (AP). Patients were treated with at least one cycle to a maximum of 6 cycles chemotherapy. Patients receiving preoperative chemotherapy were categorized in three groups (1 cycle MAP or 2 cycles AP preoperative, 2 cycles MAP or 3 cycles AP preoperative and >2 cycles MAP or >3 cycles AP preoperative). Generally, 2 cycles MAP or 3 cycles AP are used preoperatively. The other variants show patients receiving less or more cycles preoperative chemotherapy. Histopathological response on chemotherapy was obtained by a reference pathologist after histopathologic examination of the resected primary tumor. The percentage of tumor necrosis attributable to preoperative chemotherapy was defined by the Huvos grading. Huvos grading stage 1 and 2 is defined as ≤90% necrosis (bad responders). Huvos grading stage 3 and 4 defined is as >90% necrosis (good responders) [21]. 

Primary outcome was OS from surgery until death or until last date of follow-up. Secondary outcome was EFS; from resection to first event which consisted of LR, progression of metastasis, new metastasis, death or last date of follow-up. In patients with DM at presentation the next event was considered for EFS. LR was defined as a relapse of primary tumor situated at the same location of the primary tumor which was radically or marginally resected.

### 2.3. Follow-Up

Patients were followed at the outpatient clinic for local control, functional outcome and disease progression. Follow-up consisted of physical examination and radiographic control. Radiographic control comprised chest radiography and radiography of the affected bone. Follow-up visits were performed maximum 25 years after diagnosis with frequent visits in the first years after initial diagnosis and less frequent in later years according to the EURAMOS protocol [22]. 

### 2.4. Statistical Analysis

A Cox proportional hazards regression model with time fixed and time dependent covariates [23] was estimated to evaluate the association between OS, EFS and prognostic factors. Age group, location of the tumor, size of the tumor, the presence of DM at presentation, surgical margin, response to chemotherapy and local recurrence of disease were included in the Cox model. The effect of LR on survival outcomes was analyzed in two different ways, as a time-dependent covariate in the Cox model and by using the Landmark approach [24]. A landmark model only uses information available at the landmark time (t_LM_). Only patients alive at t_LM_ are included in the analysis. In our study t_LM_ is chosen at 24 months after the date of surgery. At the landmark time patients were classified as having experienced LR before 24 months or not. Survival curves were estimated using the Kaplan–Meier (KM) methodology. Outcomes were statistically significant when the *p*-value was <0.05. Because of a low number of patients for some crosstabulations, the Fisher exact test was used instead of the Chi-square test when testing categorical variables. Median follow-up time was computed using the reversed KM estimator. Missing covariates were imputed using multiple imputation methods [25] for survival data with the event indicator and the Nelson–Aalen estimator of the cumulative hazard as variables in the imputation model [26]. In total 20 data sets were imputed, Rubin’s rule was applied to obtain the final estimates along with their standard error. The analysis was performed by using SPSS (IBM Corp. Released 2017. IBM SPSS Statistics for Windows, Version 25.0. Armonk, NY: IBM Corp).

## 3. Results

### 3.1. Baseline Characteristics

The total LUMC-cohort contained 610 patients with osteosarcoma (Figure 1). Twenty patients were excluded due to secondary osteosarcoma, 88 patients due to low, intermediate or unknown grade osteosarcoma and 1 patient due to an inconclusive pathology report. Among 501 patients with high-grade osteosarcoma, 84 patients were not treated with curative intent, for 2 patients the date of resection was unknown, and 13 patients were excluded because the primary tumor was located facially or extra-skeletally (soft-tissue). After applying the exclusion criteria, 402 patients were included in this study. The median age at diagnosis was 19.14 years (range 3–82 years). The three age groups comprised 114 children (28.7%) aged 0 to <16 years, 218 (54.2%) adolescents and young adults (AYA) aged 16–<40 and 70 (17.4%) older adults aged ≥40 years. Among all patients 60% of them had a poor histopathological response on chemotherapy and 40% had a good histopathological response on chemotherapy.

### 3.2. Differences in Presentation Among Age Groups

A significant difference at presentation was found among the age groups comparing tumor location (*p* < 0.001) (Table 1). Older adults more often presented with an axial tumor compared to children and AYA. A significant difference was found among age groups and patients presenting with pathological fractures (*p* = 0.007). Of all patients, 347 (89.4%) presented without a pathological fracture of whom 102 children (90.3%), 193 AYA (92.3%) and 52 older adults (78.8%). Children were diagnosed significantly more often with DM at presentation compared to AYA and older adults (*p* = 0.037). Children, AYA and older adults, respectively, presented with at least one pulmonary metastasis in 16.5%, 12% and 5.7% of patients. Of all patients, 55 children (51.9%) underwent a radical resection compared to 99 AYA (48.3%) and 29 (42.6%) older adults. A total of 50 patients (13.2%) had an irradical resection: 7 children (6.6%), 31 AYA (15.1%) and 12 older adults (17.6%). No significant differences were found among the age groups between different types of resection (*p* = 0.070). However, the 258 patients (66.7%) receiving resection and reconstruction comprised of 77 children (71.3%), 139 AYA (66.2%), and 42 older adults (60.9%). The 56 (14.5%) patients receiving resection comprised of 7 children only (6.5%) compared to 36 AYA (17.1%) and 13 older adults (18.8%). Older adults were significantly more often treated with AP chemotherapy (*p* < 0.001), where children were more often treated with MAP (*p* < 0.001). The amount of received pre-operative chemotherapy cycles did not differ significantly among age groups. The majority of the patients (77.7%) received two MAP cycles or three AP cycles pre-operative. Finally, the response on chemotherapy differed significantly among the age groups (*p* = 0.005). Children had a good histopathological response significantly more often on pre-operative chemotherapy compared with AYA and older adults. 

### 3.3. Overall Survival in Total Cohort

Median follow-up time for the overall cohort containing 402 patients, was 136 months (95%CI 116.4–155.6). Among these patients, 5-year OS was 59.1% (95%CI 54.2–64.0). The 5-year OS for 114 children, 218 AYA and 70 older adults was, respectively, 67.2% (95%CI 58.18–76.22), 56.5% (49.84–63.16), 54.3% (42.34–66.26) as can be seen in Figure 2 and Table 3. The 5-year OS for 325 patients (83.1%) without DM at presentation was 66.1% (95%CI 60.81–71.40). OS for 66 patients (16.9%) with DM at presentation was significantly lower (*p* < 0.001) with a 5-year OS of 30% (95%CI 18.63–41.37) (Table 2, Figure 3). Among patients presenting without DM, OS differed significantly between the three age groups (*p* = 0.006). Children, AYA and older adults had, respectively, a 5-year OS of 78.5% (95%CI 87.32–69.68), 63.8% (95%CI 56.35–71.25) and 55.4% (95%CI 43.05–67.75). 

### 3.4. Event Free Survival

Of all 402 patients, 55.5% (223/402) experienced an event defined as LR, progression of metastasis, diagnosis of new metastasis or death. The 5-year EFS for 114 children, 218 AYA and 70 older adults was, respectively, 58.5% (95%CI 49.29–67.71), 40.6% (95%CI 33.94–47.26), 38.9% (95%CI 27.34–50.46) as can be seen in Table 3 and Figure 4. A total of 1, 3 and 5 years after surgery the event-free survival was, respectively, 71.6% (95%CI 67.1–76.1), 49.2% (95%CI 44.3–54.1) and 45.3% (95%CI 40.4–50.2) (Figure 5). 

### 3.5. Landmark Analysis

Survival from landmark time at 24 months post-surgery was estimated for patients with and without LR at t_LM_. In this analysis 304 patients were included; 20 patients (6.6%) had an LR within 24 months post-surgery. Patients with LR at t_LM_ had a poor survival compared to patients without (*p* < 0.001) (Figure 6).

### 3.6. Prognostic Factors

Size of the tumor (HR 1.711, 95%CI 1.193–2.455), the response to chemotherapy (HR 0.422, 95%CI 0.276–0.646), the presence of distant metastasis at presentation (HR 3.578, 95%CI 2.492–5.138) and local recurrence of disease (HR 4.456, 95%CI 2.911–6.682) were significantly associated with OS (Table 4). Age group (AYA vs. children, HR 1.499, 95%CI 1.067–2.108), (older adults vs. children, HR 1.708, 95%CI 1.094–2.666), size of the tumor (HR 1.836 95%CI 1.335–2.527), response on chemotherapy (HR 0.407, 95%CI 0.288–0.574) and distant metastasis at presentation (HR 2.575, 95%CI 1.859–3.565) were associated with EFS. Age group was found to be an independent prognostic factor of EFS but not for OS. An HR of 1.313 on OS was found comparing AYA and children (95%CI 0.891–1.935). An HR of 1.326 on OS was found comparing older adults and children (95%CI 0.802–2.193). 

## 4. Discussion

This study shows significant differences in tumor characteristics, treatment characteristics and outcome survival outcomes as OS and EFS among children, AYA and older adult population in patients with high-grade osteosarcoma. Children and AYA had better OS and EFS compared to the older adults. These results are in line with previous studies [8,11,14,15,17,18]. Older adults present more often with an axial located tumor, pathological fracture and the protocolized treatment consists more often of AP instead of MAP. Furthermore, a good histopathological response on chemotherapy is less often seen in older adults. 

In line with previous studies [3,17] age group was found to be an independent prognostic factor for EFS, resulting in poor EFS among older patients. When comparing AYA vs. children and older adults vs. children, respectively, an HR of 1.499 (95%CI 1.067–2.108) and 1.708 (95%CI 1.094–2.666) was found. A possible explanation for a poor EFS in older patients is that older patients suffer more often of axial located tumors that are technically more difficult to operate on and could lead to a higher risk of incomplete surgical resection [3,4,14,15,17]. 

A higher frequency of AP chemotherapy among an older group was possibly due to the fact that the older adults tolerate a less intensive chemotherapy protocol. Dose limitations due to comorbidities, age-related organ dysfunction or chemotherapy related toxicity might be associated to poorer response to chemotherapy compared with younger patients [8,17]. Finally, osteosarcoma in older adults seems to have another biological behavior and tends to be more resistant to chemotherapy than that in younger patients [3,8,9]. All these factors can (partly) lead to a decreased EFS in older patients.

DM at presentation is another important prognostic factor resulting in poor survival [11,27]. In this study children present more often with DM at presentation compared to AYA and older adults. Our findings are in contrast with the studies of Hagleitner et al. and Tsuda et al. [8,9], both stating that metastasis presented less frequently in younger patients. However, Hagleitner et al. and Tsuda et al. both used a different distribution of age groups (respectively, patients aged 0–14 yrs, 15–19 yrs, 20–40 yrs and patients aged <40 yrs, 41–64 yrs, >65 yrs). It is of methodological importance in which categorial variable age has been converted and therefore outcomes can vary fairly [10,17]. Another explanation could be the inclusion criteria of this study possibly resulting in a low number of older adults who are more likely to develop DM. As a result of the inclusion criteria, the number of excluded older adults with DM at presentation might be higher. Comorbidities in older adults could lead to restrictions in chemotherapy regimens and therefore have a higher risk of palliative therapy [15,28]. In the study of Tsuda et al. patients with palliative therapy were taken into account as well. In the study of Hagleitner et al. it is not clearly described if patients received palliative therapy. This led to the fact that care should be taken while comparing this study with the studies of Hagleitner et al. and Tsuda et al.

The factors associated with OS were tumor size, histopathological response to chemotherapy, DM at presentation and LR. The factors associated with an effect on EFS were age group, tumor size, histopathological response to chemotherapy and DM at presentation. These results are in line with previous studies [3,9,17,18,29,30]. Age groups were found to be an independent prognostic factor for EFS but not for OS. These results are not in line with the studies of Hagleitner et al. and Mankin et al. [8,31]. This could be explained by the fact that Hagleitner et al. performed a study with only 102 patients. Therefore, adjustment for all important variables in the multivariate analysis could not be done. Furthermore, both studies used different inclusion criteria, therefore a proper comparison could not be made. Finally, care should be taken when interpreting the effect of histopathological response on OS and EFS. In the multivariate analysis, both AP as MAP chemotherapy were taken into account while analyzing the effect on histopathological response. The histopathological response in patients receiving AP chemotherapy is evaluated earlier (after 6 weeks) in comparison to patients receiving MAP (after 10 weeks). In addition, MAP is a more intensive chemotherapy regimen compared to AP and therefore possibly influencing the effect on the primary outcome. 

After 40 years of (neo)adjuvant chemotherapy for osteosarcoma, whose benefits in terms of survival are well established but have not improved, this paper clearly shows that it is time to change the approach and consider additional therapeutic options. In recent years there have been no major results in phase 3 trials in the (neo)adjuvant treatment of patients with resectable osteosarcoma. Phase-2 trials so far have shown no effective trials for poor prognosis osteosarcoma [32,33,34]. The international community of physicians involved in this disease awaits results of the investigation of the complete genomic landscape of osteosarcoma [35]. Insights from pan-genomic studies could gain a better insight into the development and clonal evolution of this malignancy, that hopefully will lead to the development of more specific drugs for osteosarcoma [36].These results should guide the development of new (neo)adjuvant trials.

### Strengths and Limitations

Our study is one of the largest single center studies investigating prognostic factors on survival. This cohort offers a long median follow-up time of 136 months. In addition, it is one of the few studies describing patient and treatment characteristics in three different age groups and therefore it could be directive to future studies. Other studies describe small study populations or present data from prospective or randomized controlled trials with different pre-empted endpoints and inclusion criteria [8,9,14,15,16,17,18].

Due to the retrospective nature of this study, several limitations were present. In this study we were unable to assess histopathological response per type of chemotherapy regimen. Although histopathological response is an important prognostic factor influencing OS and EFS, care should be taken be taken while interpreting these data. Furthermore, we were unable to assess the association of chemotherapy treatment with survival in the multivariate analysis. Finally, not all known pathological and biochemical features of osteosarcoma patients were taken into account in this paper. The retrospective nature of this study explains for the lack of some possibly important prognostic factors that could not be retrieved for most of the patients. 

## 5. Conclusions

In this single center study, we found poor OS and EFS in older adults with high-grade osteosarcoma compared to AYA and children. Large tumor size, a poor histopathological response, DM at presentation and LR are important independent prognostic factors influencing OS negatively. Age group (older adults), large tumor size, a poor histopathological response and DM at presentation were found to be important independent prognostic factors influencing EFS negatively. DM and LR can make a significant difference in prognosis and is therefore key in the approach of patients suffering high-grade skeletal osteosarcoma. Differences in outcome among different age groups can be partially explained by patient and treatment characteristics. 

## Figures and Tables

**Figure 1 cancers-13-00486-f001:**
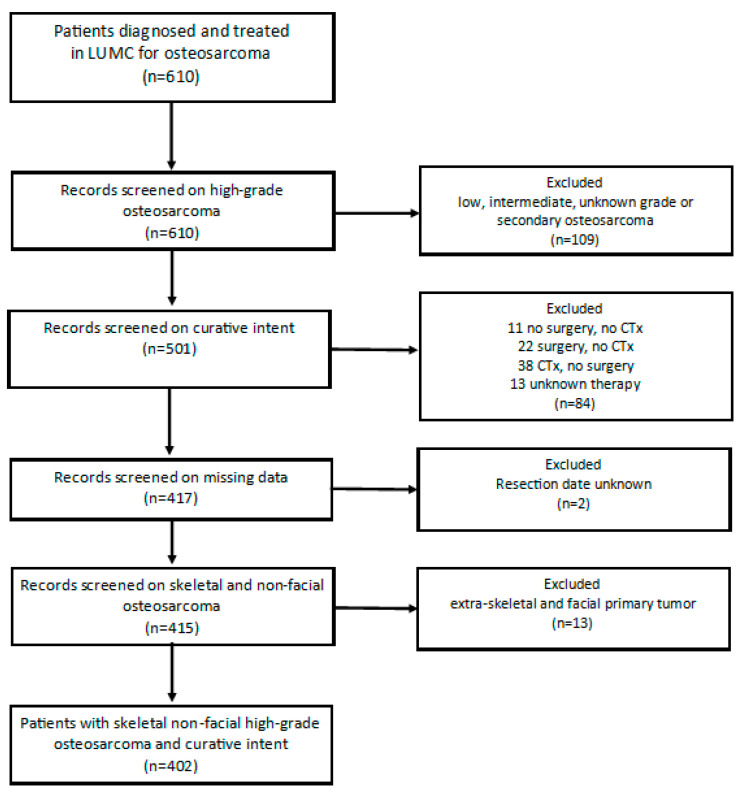
Flowchart patient selection. Legend; LUMC = Leiden University Medical Center, CTx = Chemotherapy.

**Figure 2 cancers-13-00486-f002:**
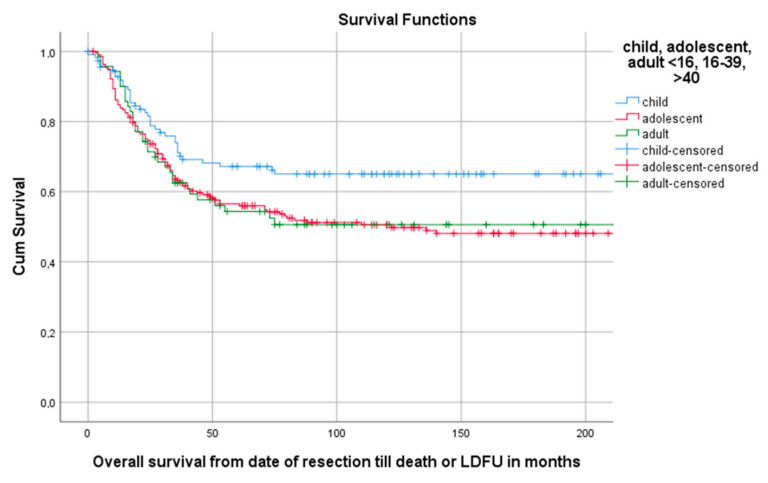
Kaplan–Meier (KM) estimation of OS in the total cohort divided by age group. Legend: OS = overall survival, cum survival = cumulative survival, LDFU = last date of follow-up.

**Figure 3 cancers-13-00486-f003:**
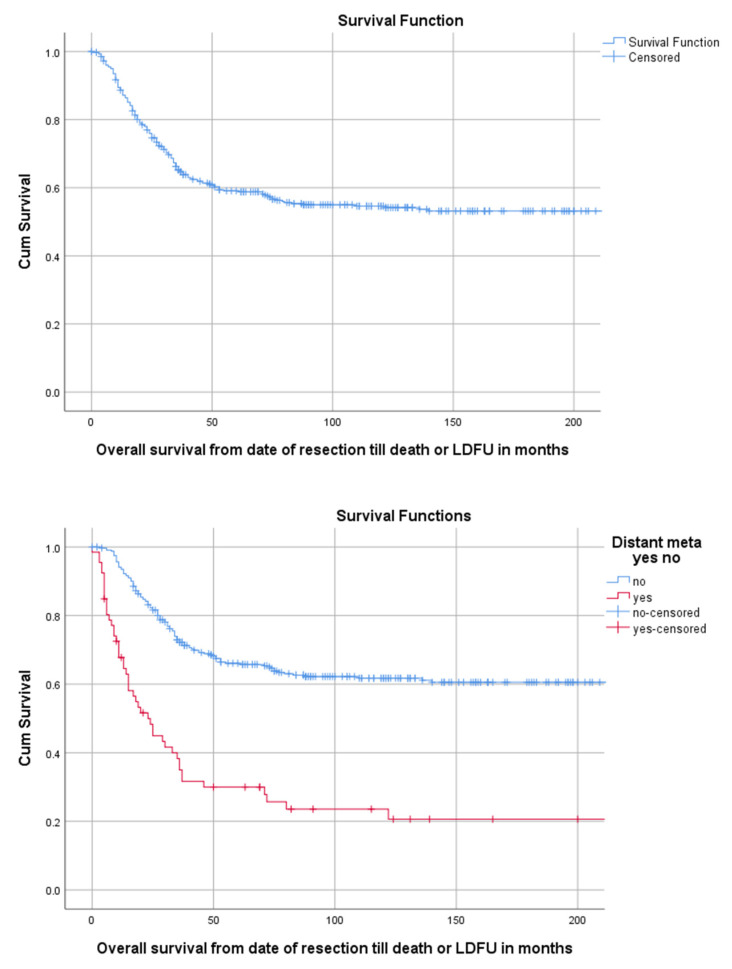
KM estimation of OS in the total cohort (upper panel) and of patients with and without distant metastasis (lower panel). Legend: OS = overall survival, DM = distant metastasis at presentation, cum survival = cumulative survival, LDFU = last date of follow-up.

**Figure 4 cancers-13-00486-f004:**
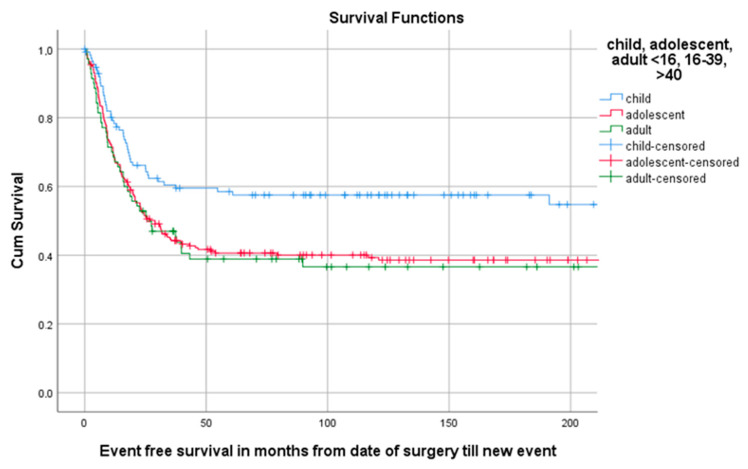
KM estimation of EFS in total cohort divided by age group. Legend: EFS = event-free survival, cum survival = cumulative survival.

**Figure 5 cancers-13-00486-f005:**
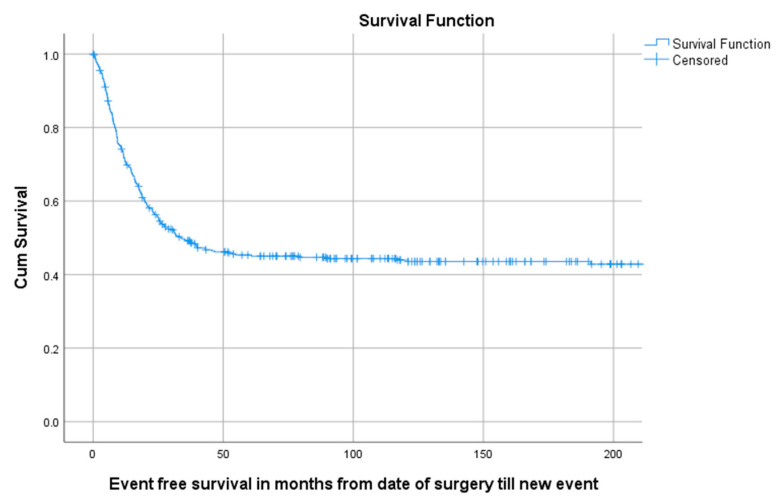
KM estimation of EFS of total cohort. Legend: EFS = Event-free survival, cum survival = cumulative survival.

**Figure 6 cancers-13-00486-f006:**
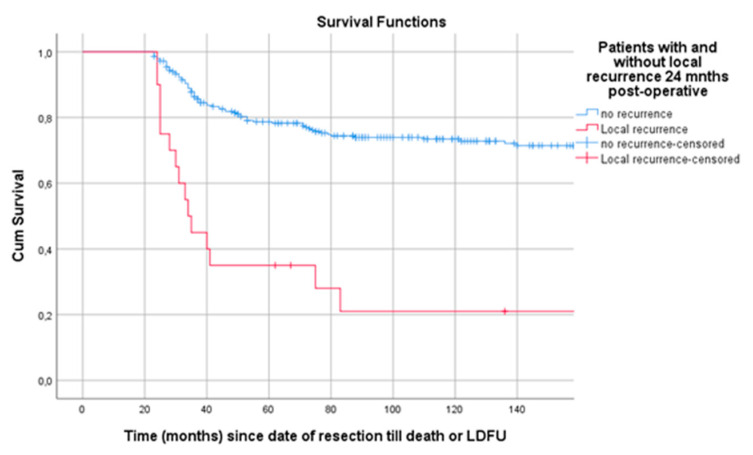
Landmark analysis of patients with and without LR 24 months post-surgery. Legend: LR = local recurrence, cum survival = cumulative survival, LDFU = last date of follow-up.

**Table 1 cancers-13-00486-t001:** Characteristics of the overall cohort diagnosed with skeletal high-grade osteosarcoma.

Characteristic	N (%)	Children (0–<16 yrs)	AYA (16–<40 yrs)	Older Adults (≥40 yrs)	*p*-Value
**Gender**	402	114 (28.7)	218 (54.2)	70 (17.4)	0.092
Male	228 (56.7)	64 (56.1)	132 (57.9)	32 (45.7)	
Female	174 (43.3)	50 (43.9)	86 (39.4)	38 (54.3)	
**Location tumor**	402	114 (28.4)	218 (54.2)	70 (17.4)	<0.001
Extremities	372 (92.5)	112 (98.2)	203 (93.1)	57 (81.4)	
Axial (pelvis, chest, spine)	30 (7.5)	2 (1.8)	15 (6.9)	13 (18.6)	
**Tumor size**	375	107 (28.5)	200 (53.3)	68 (18.1)	0.377
Small (≤8 cm)	154 (41.1)	43 (40.2)	78 (39)	33 (48.5)	
Large (≥8 cm)	221 (58.9)	64 (59.8)	122 (61)	35 (51.5)	
**Pathologic fracture**	388	113 (29.1)	209 (53.9)	66 (17)	0.007
No	347 (89.4)	102 (90.3)	193 (92.3)	52 (78.8)	
Yes	41 (10.6)	11 (9.7)	16 (7.7)	14 (21.2)	
**Distant metastasis at presentation**	391	111 (28.4)	210 (53.7)	70 (17.9)	0.037
No	325 (83.1)	87 (78.4)	173 (82.4)	65 (92.9)	
Yes	66 (16.9)	24 (21.6)	37 (17.6)	5 (7.1)	
***No. of lungmets at presentation**	388	109 (28.1)	209 (53.9)	70 (18)	0.389
None	341 (87.9)	91 (83.5)	184 (88)	66 (94.3)	
1	9 (2.3)	3 (2.8)	6 (2.9)	0 (0)	
2–5	30 (7.7)	11 (10.1)	16 (7.7)	3 (4.3)	
>5	8 (2.1)	4 (3.7)	3 (1.4)	1 (1.4)	
**Surgical margin**	379	106 (28)	205 (54.1)	68 (17.9)	0.178
Radical	183 (48.3)	55 (51.9)	99 (48.3)	29 (42.6)	
Marginal	146 (38.5)	44 (41.5)	75 (36.6)	27 (39.7)	
Irradical	50 (13.2)	7 (6.6)	31 (15.1)	12 (17.6)	
**Type of resection**	387	108 (27.9)	210 (54.3)	69 (17.8)	0.070
Resection/reconstruction	258 (66.7)	77 (71.3)	139 (66.2)	42 (60.9)	
Amputation/exarticulation	73 (18.9)	24 (22.2)	35 (16.7)	14 (20.3)	
Resection only	56 (14.5)	7 (6.5)	36 (17.1)	13 (18.8)	
**Chemotherapy treatment**	359	98 (27.3)	198 (55.6)	63 (17.5)	<0.001
Intention AP	225 (62.7)	43 (43.9)	125 (63.1)	57 (90.5)	
Intention MAP	134 (37.3)	55 (56.1)	73 (36.9)	6 (9.5)	
***Pre-op CTx cycles**	309	89 (28.8)	176 (57)	44 (14.2)	0.256
1 MAP or 2 AP	41 (13.3)	12 (13.5)	22 (12.5)	7 (15.9)	
2 MAP or 3 AP	240 (77.7)	74 (83.1)	134 (76.1)	32 (72.7)	
>2 MAP or >3 AP	28 (9.1)	3 (3.4)	20 (11.4)	5 (11.4)	
***Response on chemotherapy**	337	105 (31.2)	184 (54.6)	48 (14.2)	0.005
Poor (Huvos 1,2)	202 (59.9)	51 (48.6)	115 (62.5)	36 (75)	
Good (Huvos 3,4)	135 (40.1)	54 (51.4)	69 (37.5)	12 (25)	
***/** Local recurrence**	391	106 (27.1)	215 (55)	70 (17.9)	
No	346 (88.5)	102 (96.2)	190 (88.4)	54 (77.1)	
Yes	45 (11.5)	4 (3.8)	25 (11.6)	16 (22.9)	

Legend: AYA = Adolescent and Young Adult, Lungmets = lung metastasis, AP = Adriamycine-CisPlatin, MAP = Methotrexate-Adriamycine-CisPlatin, CTx = Chemotherapy, pre-op = pre-operative, * Fisher exact test because number of patients <5, ** No *p*-value because of time dependent variable.

**Table 2 cancers-13-00486-t002:** Overall survival (OS) among different age groups with or without distant metastasis (DM) at presentation.

Factors	N (%)	5-yr OS among M0 (%)	*p*-Value	N (%)	5-yr OS among M1 (%)	*p*-Value
**Overall group**	325 (83.1)	66.1		66 (16.9)	30	
			0.006			0.971
Child (0–<16)	87 (26.8)	78.5		24 (36.4)	21.7	
AYA (16–<40)	173 (53.2)	63.8		37 (56.1)	32.4	
Older adults ≥40	65 (20)	55.4		5 (7.6)	40	

Legend: M0 = patients without metastasis at presentation, M1 = patients with metastasis at presentation.

**Table 3 cancers-13-00486-t003:** OS and EFS at 5 years along with 95% confidence interval (CI).

Factors	N (%)	5-Year OS (%) with 95%CI	*p*-Value	N (%)	5-Year EFS (%) with 95%CI	*p*-Value
**Sex**	402		0.126	402		0.033
Male	228 (56.7)	55.5 (48.8–62.16)		228 (56.7)	40.7 (34.23–47.17)	
Female	174 (43.3)	63.6 (56.35–70.85)		174 (43.3)	51.3 (43.85–58.75)	
**Age group**	402		0.044	402		0.007
Child (0–<16)	114 (28.4)	67.2 (58.18–76.22)		114 (28.4)	58.5 (49.29–67.71)	
AYA (16–<40)	218 (54.2)	56.5 (49.84–63.16)		218 (54.2)	40.6 (33.94–47.26)	
Older adults ≥40	70 (17.4)	54.3 (42.34–66.26)		70 (17.4)	38.9 (27.34–50.46)	
**Location**	402		0.960	402		0.361
Extremities	372 (92.5)	59.1 (54.0–64.2)		372 (92.5)	45.8 (40.70–50.90)	
Axial (chest, spine, pelvis)	30 (7.5)	60 (42.56–77.44)		30 (7.5)	40 (22.56–57.44)	
**Tumor size**	375		<0.001	375		<0.001
Small ≤8 cm	154 (41.1)	72.4 (65.15–79.65)		154 (41.1)	70.1 (52.26–67.94)	
Large ≥8 cm	221 (58.9)	50.2 (43.34–57.06)		221 (58.9)	34.5 (28.03–40.97)	
**Surgical margin**	379		0.037	379		0.030
Radical	183 (48.3)	60.7 (53.45–67.95)		183 (48.3)	48.2 (40.75–55.65)	
Marginal	146 (38.5)	62.3 (54.26–70.34)		146 (38.5)	47.5 (39.27–55.73)	
Irradical	50 (13.2)	45.4 (31.48–59.32)		50 (13.2)	29.9 (17.16–42.64)	
**Type of resection**	387		0.002	387		0.004
Resection/reconstruction	258 (66.7)	60.6 (54.52–66.68)		258 (66.7)	47.1 (40.83–53.37)	
Amputation/exarticulation	73 (18.9)	45.7 (34.14–57.26)		73 (18.9)	33.6 (22.62–44.58)	
Resection only	56 (14.5)	72.2 (60.24–84.16)		56 (14.5)	56.7 (43.57–69.83)	
**Response on chemotherapy**	337		<0.001	337		<0.001
Poor (Huvos 1,2)	202 (59.9)	46.6 (39.54–53.66)		202 (59.9)	31.2 (24.73–37.67)	
Good (Huvos 3,4)	135 (40.1)	74.5 (67.05–81.95)		135 (40.1)	66.9 (58.86–74.94)	
**Distant metastasis at presentation**	391		<0.001	391		<0.001
No	325 (83.1)	66.1 (60.81–71.39)		325 (83.1)	50.9 (45.41–56.39)	
Yes	66 (16.9)	30 (18.63–41.37)		66 (16.9)	20.9 (10.71–31.09)	

Legend: CTx = Chemotherapy.

**Table 4 cancers-13-00486-t004:** Hazard ratio for prognostic factors on OS and EFS along with the 95% confidence interval estimated with the Cox proportional hazards regression model.

Factors	HR_OS_	95% CI	*p*-Value	HR_EFS_	95% CI	*p*-Value
**Sex**			0.490			0.097
Male						
Female	0.891	0.642–1.237		0.786	0.592–1.044	
**Age group**						
Child (0–<16)	*Reference group*			*Reference group*		
AYA (16–<40)	1.313	0.891–1.935	0.168	1.499	1.067–2.108	0.020
Older adults ≥40	1.326	0.802–2.193	0.272	1.708	1.094–2.666	0.018
**Location**			0.678			0.346
Extremities						
Axial (chest, spine, pelvis)	0.868	0.446–1.692		1.277	0.768–2.123	
**Tumor size**			0.004			<0.001
Small ≤8 cm						
Large ≥8 cm	1.711	1.193–2.455		1.836	1.335–2.527	
**Surgical margin**						
Radical	*Reference group*			*Reference group*		
Marginal	0.839	0.586–1.203	0.340	0.941	0.689–1.285	0.702
Irradical	1.248	0.783–1.988	0.351	1.141	0.769–1.693	0.513
**Response on chemotherapy**			<0.001			<0.001
Poor (Huvos 1,2)						
Good (Huvos 3,4)	0.422	0.276–0.646		0.407	0.288–0.574	
**Distant metastasis at presentation**			<0.001			<0.001
No						
Yes	3.578	2.492–5.138		2.575	1.859–3.565	
**** Local recurrence**			<0.001			
No						
Yes	4.456	2.911–6.682				

Legend: CTx = Chemotherapy, ** = *time dependent variable*, *HR* = *Hazard Ratio*.

## Data Availability

The data presented in this study are available on request from the corresponding author. The data are not publicly available since the database consists of single center data and other studies are currently being conducted with this data.

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
