# Peer review of "Survival Analysis of 3 Different Age Groups and Prognostic Factors among 402 Patients with Skeletal High-Grade Osteosarcoma. Real World Data from a Single Tertiary Sarcoma Center"

_cancers, 2021, doi:10.3390/cancers13030486_

Round 1

Reviewer 1 Report

This paper provides a reliable description of the clinical characteristics of high grade OS treated in a single centre and confirms some well known prognostic factors: age, size, mets at clinical onset. I wonder how much does this contribution contributes to a better understanding of the disease and, in turn, where is the novelty of this analysis in comparison to similar reports?

Besides, this study does not include other valuable clinical, pathological and biochemical features that are also prognostic, therefore is only partially descriptive of the elements that, all together combined, might help the physician to draw patient-adjusted regimens of therapy. In that sense, I am afraid that it is incomplete.

After 40 years of neoadjuvant chemo for OS, whose benefits in terms of survival are well established but have not improved, this paper clearly shows that it is time to change the approach and consider additional therapeutic options. I would like to ask the authors to express their valuable opinion and provide some suggestions. 

Author Response

Thank you very much for your valuable comments and your critical review of our paper. Although no ground-breaking new therapies follow from our paper, we think this paper is of significant importance as it provides long term follow up from a large real world patient cohort of a tertiary sarcoma center and, as you state it; clearly underlines the need for additional therapeutic and individualized options for osteosarcoma patients. In addition the performed sophisticated statistical analysis underline the differences in outcome for individual patients’ perspectives.

We agree with your observation that not all known pathological and biochemical features of osteosarcoma patients were taken into account in this paper. The retrospective nature of this study explains for the lack of some possibly important prognostic factors that could not be retrieved for most of the patients. However, we think most of the important clinical prognostic factors were taken into account in our analysis. Once again, we agree with you, therefore we added this as a limitation in the discussion section. See line 351-354.

“After 40 years of neoadjuvant chemo for OS, whose benefits in terms of survival are well established but have not improved, this paper clearly shows that it is time to change the approach and consider additional therapeutic options.” We thankfully agree with this comment and have added its content in to the discussion paragraph: See line 328-330.

I would like to ask the authors to express their valuable opinion and provide some suggestions. 

In order to respond on your comment we added the following text to the discussion section. See line 330-338.

In recent years there have been no major results in phase 3 trials in the (neo)adjuvant treatment of patients with resectable osteosarcoma. Phase-2 trials so far have shown no effective trials for poor prognosis osteosarcoma (van Maldegem AM et al Clin Sarcoma Res 2012; PMID 22587841; Lagmay LP et al. J clin Oncol 2016, 34: PMID 27400942; Omer N. et al Eur J Cancer 2017, PMID 28219023). The international community of physicians involved in this disease awaits results of the investigation of the complete genomic landscape of osteosarcoma (Roberts RD et al Cancer 2019,PMID 31355930). Insights from pan-genomic studies could gain a better insight in the development and clonal evolution of this malignancy, that hopefully will lead to the development of more specific drugs for osteosarcoma (Tirtei E et al. Pediatr Blood Cancer 2019, PMID 31736201). These results should guide the development of new (neo)adjuvant trials.

Reviewer 2 Report

In the manuscript by Evenhuis et al., titled, "Survival analysis of 3 different age groups and prognostic factors among 402 patients with skeletal high-grade osteosarcoma. Real world data from a single tertiary sarcoma center" the authors present another retrospective study on osteosarcoma survival. The research team is very experienced and has been publishing similar data and analyses for the past decade and are very capable and thorough in their approach. However, there is no ground-breaking data or conclusions that change the paradigm in how we think about the treatment and survival of patients with osteosarcoma. Patients with metastasis have a more dismal prognosis as do patients with early recurrence within the landmark time of 24 months. 

The curated patient data was assembled well with logical exclusion and inclusion criteria. The strength of the data coming from a single center minimizes some of the variables associated with multicenter studies. 

Minimal suggestions for improvement include changing the title of the x-axis of figure 3 to something more informative and less misleading. 

Also on line 170 of page 5, the authors state that a "remarkable difference" was observed between age groups and the type of resection, when the p value provided is > 0.05. In the methods section they state that only p values < 0.05 would be considered significant. Even though they do not use the word significant, remarkable is very similar and would encourage the authors to reword their statement. 

Author Response

Thank you very much for your valuable comments and your critical view on our paper.
Although no ground-breaking new therapies follow from our paper, we think this paper is of significant importance as it provides long term follow up from a large real world patient cohort of a tertiary sarcoma center and it clearly underlines the need for additional therapeutic and individualized options for osteosarcoma patients. In addition the performed sophisticated statistical analysis underline the differences in outcome for individual patients’ perspectives. As you state, it  is known that patients with distant metastasis and/or local recurrence have dismal prognosis. However, limited data and studies correctly describing the effect of LR on survival are available. This paper describes the effect of LR and other prognostic factors on OS based on a sophisticated statistical analysis and on a large number of patients with long-term follow-up.  

The curated patient data was assembled well with logical exclusion and inclusion criteria. The strength of the data coming from a single center minimizes some of the variables associated with multicenter studies. 

Thank you for this kind comment

Minimal suggestions for improvement include changing the title of the x-axis of figure 3 to something more informative and less misleading. 

The x-axis of figure 3 (currently figure 5) was adjusted to “Event free survival in months from date of surgery till new event”. See line 240.

Also on line 170 of page 5, the authors state that a "remarkable difference" was observed between age groups and the type of resection, when the p value provided is > 0.05. In the methods section they state that only p values < 0.05 would be considered significant. Even though they do not use the word significant, remarkable is very similar and would encourage the authors to reword their statement. 

You are absolutely right about the cut-off value for significance. We used the wrong word in sentence on line 170 of page 5. Therefore we adjusted it to: No significant differences were found among the age groups between different types of resection (p=0.070). However, the 258 patients (66.7%) receiving resection and reconstruction comprised of 77 children (71.3%), 139 AYA (66.2%), and 42 older adults (60.9%). See line 171-174.

Reviewer 3 Report

Authors reported age is a strong prognosis factor for high grade skeletal osteosarcoma. OS of young tumors responded better and overall survival was better compared to older patients. The manuscript is relatively simple but easy to read. I enjoyed reviewing this manuscript. I don't have any major comment for this manuscript but I recommend authors should add Kaplan Meier curves of three ages for survival. This is one of major findings in this manuscript.

Round 2

Reviewer 1 Report

Thnaks for having positively considered my previous comments.